# The Kinesin Gene *KIF26B* Modulates the Severity of Post-Traumatic Heterotopic Ossification

**DOI:** 10.3390/ijms23169203

**Published:** 2022-08-16

**Authors:** George A. E. Pickering, Favour Felix-Ilemhenbhio, Matthew J. Clark, Klaudia Kocsy, Jonathan Simpson, Ilaria Bellantuono, Alison Gartland, Jeremy Mark Wilkinson, Konstantinos Hatzikotoulas, Endre Kiss-Toth

**Affiliations:** 1Department of Oncology & Metabolism, University of Sheffield, Sheffield S10 2TN, UK; 2Healthy Lifespan Institute, University of Sheffield, Sheffield S10 2TN, UK; 3Department of Infection, Immunity and Cardiovascular Disease, University of Sheffield, Sheffield S10 2TN, UK; 4Institute of Translational Genomics, Helmholtz Zentrum München—German Research Center for Health and the Environment, 85764 Neuherberg, Germany

**Keywords:** heterotopic ossification, severity, risk, KIF26B, CRISPR-Cas9

## Abstract

The formation of pathological bone deposits within soft tissues, termed heterotopic ossification (HO), is common after trauma. However, the severity of HO formation varies substantially between individuals, from relatively isolated small bone islands through to extensive soft tissue replacement by bone giving rise to debilitating symptoms. The aim of this study was to identify novel candidate therapeutic molecular targets for severe HO. We conducted a genome-wide scan in men and women with HO of varying severity following hip replacement for osteoarthritis. HO severity was dichotomized as mild or severe, and association analysis was performed with adjustment for age and sex. We next confirmed expression of the gene encoded by the lead signal in human bone and in primary human mesenchymal stem cells. We then examined the effect of gene knockout in a murine model of osseous trans-differentiation, and finally we explored transcription factor phosphorylation in key pathways perturbed by the gene. Ten independent signals were suggestively associated with HO severity, with *KIF26B* as the lead. We subsequently confirmed *KIF26B* expression in human bone and upregulation upon BMP2-induced osteogenic differentiation in primary human mesenchymal stem cells, and also in a rat tendo-Achilles model of post-traumatic HO. CRISPR-Cas9 mediated knockout of *Kif26b* inhibited BMP2-induced *Runx2*, *Sp7/Osterix*, *Col1A1*, *Alp*, and *Bglap*/*Osteocalcin* expression and mineralized nodule formation in a murine myocyte model of osteogenic trans-differentiation. Finally, KIF26B deficiency inhibited ERK MAP kinase activation during osteogenesis, whilst augmenting p38 and SMAD 1/5/8 phosphorylation. Taken together, these data suggest a role for KIF26B in modulating the severity of post-traumatic HO and provide a potential novel avenue for therapeutic translation.

## 1. Introduction

The term heterotopic ossification (HO) describes the formation of bone at non-skeletal sites, and is a common complication of trauma, including blast injury and pelvic surgery [1,2,3,4]. HO also occurs after traumatic brain injury and burns [5,6]. The severity and clinical impacts of post-traumatic HO very dramatically between individuals [7], from small islands of HO within the soft tissues that are clinically silent, through to large lesions causing pain, swelling, restricted joint movement and rarely, joint ankylosis. Male sex, hypertrophic osteoarthritis, ankylosing spondylitis, and black ethnicity are risk factors for HO development after hip replacement, and indicate a common, complex etiology [8,9]. Initiation of a new HO lesion involves tissue injury that results in a signal to initiate endochondral or intra-membranous ossification. The dominant cell types in HO include FAPs, endothelial cells, hematopoietic cells, tendon and ligament progenitor cells, pericytes and Hoxa11+ mesenchymal stromal cells. The dominant pathways in HO include BMP, mTOR and RAR signalling. A more detailed overview of our current understanding of the molecular biology of HO is given in a recent review in this journal [10].

Current treatment approaches for HO include local irradiation and non-steroidal anti-inflammatory agents, both of which have significant side-effects [11,12]. Surgical excision is the only treatment for established HO, and is associated with a significant risk of recurrence [7,13]. Whilst the blockade of ALK2 signaling through ligand binding, inhibition of ALK2 kinase activity, or upregulation of the expression of ACVR1 at the transcriptional level have been explored in experimental models of fibrodysplasia ossificans progressiva (a rare heritable form of HO), retinoid use has off-target effects and the mechanistic justification for their use in post-traumatic HO is not established [14].

Given that disease severity is a key factor in determining the requirement for prophylactic intervention, we used hip replacement as a human model for post-traumatic HO, as a broad range of HO severity phenotypes may be captured. Here, we used an exploratory genome-wide association analysis (GWAS) as a screening approach to identify possible molecular signals for disease severity [15]. We subsequently followed up the putative lead candidate in human and animal models to determine the effect of its modulation on osseous trans-differentiation.

## 2. Results

### 2.1. Screening GWAS

Although this cohort represented the largest of HO positive phenotyped and genotyped patients, the sample size remained limited for a GWAS discovery population. As anticipated given the limited sample size, we did not identify any genome-wide significant variants associated with HO severity (Figure 1A and Appendix A). However, 10 independent signals with clumped index variants at *p* < 5 × 10^−6^ were suggestively associated with disease severity (Appendix A). The lead signal lay within an intronic region of *KIF26B* (rs35338958, EA T, EAF 0.10, OR 3.04 [1.85–5.01], *p* = 1.65 × 10^−6^; Figure 1B). *KIF26B* encodes KIF26B, a kinesin superfamily protein and downstream target of the Wnt5a-Ror and VEGF signaling pathways that plays a role in tissue morphogenesis [16,17]. Independent investigators have found upregulation of KIF26B in a mouse model of ectopic calcification at the knee [18], and intronic variation within KIF26B has been associated with bone size (http://mskkp.org/gene/geneInfo/KIF26B, accessed on 1 July 2022). Given the suggestive association identified in our screening of GWAS and the previous literature supporting a role of KIF26B in bone biology, this gene was carried forward as the candidate locus for subsequent functional analyses.

### 2.2. KIF26B Is Expressed in Human Bone, in BMP2 Stimulated Human Mesenchymal Stem Cells, Co-Localises with Microtubules in the Cytoskeleton, and Is Induced in a Mammalian Post-traumatic HO Model

Total RNA was extracted from fresh frozen, surgically excised bone from patients undergoing joint replacement. *KIF26B* expression was confirmed by real time quantitative polymerase chain reaction (RT-qPCR, Figure 2A). Next, we demonstrated *KIF26B* differential expression in human multipotent adipose-derived stem cells (hMAD, Figure 2B) [19] and in primary human bone marrow-derived mesenchymal stem cells (hMSCs, Figure 2C) in response to stimulation with BMP2. In silico analysis of a published RNASeq dataset in an in vivo model of ectopic bone formation using human fetal MSCs from an independent study [20] also demonstrated *KIF26B* induction during osteogenesis (data not shown). In addition to upregulation of *KIF26B* expression, KIF26B protein was also robustly expressed in hMSCs after osteogenic differentiation, and was partly co-localized with the cytoskeleton in MSCs (Figure 2D). Figure 2E shows secondary negative control.

To study the in vivo expression of KIF26B protein following injury, we used immunohistochemistry to identify and co-localize KIF26B expression with HO formation in tissue from a rat tendo-Achilles scalpel-induced injury model [21,22]. In these tissues taken at 10 weeks after injury, islands of new bone formation were identified and co-localized with KIF26B expression (Figure 2F).

### 2.3. Knockout of Murine Kif26b Prevents BMP2-Mediated Mineralisation of C2C12 Myoblasts

Whilst in vitro models that precisely recapitulate the injury and cellular pathogenesis of clinical HO are lacking, C2C12 mouse myoblasts are a commonly used proxy [23,24]. To substantiate the functional impact of KIF26B during osteogenic trans-differentiation, C2C12 mouse myoblasts were cultured following BMP2 stimulation, leading to a robust increase in *Kif26b* RNA from day 8 (Figure 3A). Of note, *Kif26b* was very low or completely absent in undifferentiated cells at day 0. This absence of expression was further verified at the protein level, as was its upregulation after 8 days of osteogenic differentiation. *Kif26b* was only expressed in the presence of BMP2 plus osteogenic supplements and was not detected with either low serum media, or with osteogenic supplements alone.

In cells engineered with CRISPR/Cas9 to not produce KIF26B, there was no expression of *Kif26b* RNA with BMP2 stimulation, in contrast to that observed in the wild-type C2C12 myoblasts (Figure 3B). In the *Kif26b^CRISPR^* cells, mRNA expression was close to the limit of detection at all time-points studied. Western blot confirmed that no KIF26B protein was produced by the *Kif26b^CRISPR^* cells after 8 days of BMP2 stimulation (Figure 3C). Cells of different passage number were stimulated with BMP2 in independent experiments and lysed for Western blotting (n = 3), confirming the stability of the *Kif26b^CRISPR^* in the C2C12 myoblasts. Very little mineral (Alizarin Red S staining) was produced by day 8 in either the WT or *Kif26b^CRISPR^* cells. By day 24, only a small amount of mineral was produced by the *Kif26b^CRISPR^* cells whilst the WT cells produced abundant mineral (3.7% vs. 43.3% mean mineralization, Figure 3D,E).

### 2.4. Kif26b Knockout Disrupts the Mineralisation and Trans-Differentiation of C2C12 Myoblasts by Modulating the Expression of Osteogenic Genes

Given the lack of mineralization in the *Kif26b^CRISPR^* cells, we hypothesized that the expression of genes driving osteogenesis would be abrogated. RT-qPCR analysis of *Runx2*, *Osx* and osteocalcin (*Bglap*) at day 0 (confluency), day 8, 16 and 24 was conducted to test this. At day 0, there was no difference in target gene expression between WT and *Kif6b^CRISPR^*. The greatest differential gene expression between cell types was observed at day 8 (Figure 3F). Expression of *Runx2* peaked at day 8 in the WT cells (WT versus *Kif26b^CRISPR^* cells, *p* = 0.0052). *Osx* expression in WT cells increased relative to the *Kif26b^CRISPR^* cells at days 8, 16 and 24 (*p* < 0.0001 at all treatment timepoints). We also found that *Bglap* was upregulated in the WT cells but not expressed in *Kif26b^CRISPR^* cells at days 8, 16 and 24 (Figure 3F; *p* < 0.0001).

### 2.5. KIF26B Deficiency Inhibits ERK MAP Kinase Activation during Osteogenesis, Whilst Augmenting p38 and SMAD 1/5/8 Phosphorylation

In order to understand the molecular mechanism for the abrogated osteogenic response to BMP2 stimulation in *Kif26b^CRISPR^* cells, the activity of key BMP2-effector pathways was tested in wild type and *Kif26b^CRISPR^* cells before and after osteogenic differentiation (Figure 4A,B). The activation of ERK MAPK is critical to osteoblast differentiation in mouse models [25], *Runx2* and osteocalcin/*Bglap* being major targets of this pathway [26,27,28]. In line with these studies, we found that phosphorylation of ERK during osteogenesis was profoundly inhibited in the absence of KIF26B protein (Figure 4A,B). At the same time, phosphorylation of SMAD1/5 and p38 MAPK were not negatively affected in *Kif26b^CRISPR^* cells, suggesting that *Kif26b* is not a critical regulator of these signaling axes in our BMP2-driven model. Mechanistically, RUNX2 has been shown to co-localize with SMADs and form part of a transcriptional activator complex to initiate BMP2-induced osteogenesis [29]. We thus speculate that the impaired *Runx2* expression seen in *Kif26b^CRISPR^* cells may be sufficient to inhibit osteogenesis. A summary of the proposed functional mechanisms for KIF26B is shown in Figure 4C.

## 3. Discussion

We used cohort clinical and genotype data to conduct an exploratory GWAS screen of common genetic variants to identify potential molecular candidates for the modulation of HO severity. Given the limited sample size of the GWAS, we chose *p* < 5 × 10^−6^ as the suggestive screening threshold for further analysis, and considered the GWAS data as indicative, rather than definitive. The approach of using a discovery-only GWAS (at a less conservative *p* < 5 × 10^−5^ than reported here) as supportive evidence has been successfully used by other investigators in the setting of bone biology [30,31], and in other human traits [32,33]. These analyses helped to establish the functional role of associated target genes in human biology rather than validate the causal association of an individual SNV. Our analysis, supported by bioinformatics data and previous literature, prioritized the gene encoding KIF26B as a candidate target for further functional investigation. This finding, that has recently been independently recapitulated in a murine model of intra-articular calcification, showed that *KIF26b* silencing prevents osseous trans-differentiation of progenitor cells in this setting [34]. We do not claim that the lead SNV is functional, but rather that it supports further exploration of the gene within which it resides as potentially of functional relevance in HO severity. Our subsequent analyses confirmed *KIF26B* expression in human bone, differential expression in response to a bone morphogenetic stimulus in primary human MSCs retrieved from both adults and children, and co-localization of KIF26B protein to cytoskeletal microtubules during osteogenic differentiation. The presence of KIF26B protein was also confirmed in a rat tendo-Achilles model of HO. We finally showed that modulation of Kif26b in a murine myoblast model is sufficient to inhibit osseous trans-differentiation, an effect that is associated with ERK1/2 dysregulation.

We used BMP2 as a standardized pro-osteogenic stimulus, as BMP2 is overexpressed in clinically evolving HO tissue after trauma [35,36], and BMP type 1 or 2 receptor inhibition reduces HO formation in experimental models [37,38]. BMP signaling is transduced via p38 mitogen-activated protein kinase (MAPK) [39] and/or SMAD complexes [40]. The canonical *Wnt*/ß-catenin pathway is also implicated in chondrocyte maturation and osteoblast differentiation [41,42]. A recent study showed that TNF inhibited BMP2-mediated osteogenesis via a strong activation of p38 MAPK in C2C12 and MC3T3-E1 cells whilst inhibition of p38 rescued the expression of RUNX2 and other osteogenic markers induced by BMP2 [43]. Our analysis of the consequences of *Kif26b*-deficiency demonstrated over-activity of this pathway, which may also contribute to the observed lack of osteogenesis.

Our data suggested that KIF26B inhibition might have an investigational therapeutic role in post-traumatic HO prevention. Our immunohistochemistry data demonstrated co-localization of KIF26B to microtubules during osseous trans-differentiation, consistent with its motor function [44,45]. KIF motor properties are currently being explored for therapeutic benefit in cancer clinical trials, including KIF11 inhibitors SB-743921 (Cytokinetics, San Francisco, CA, USA), ARRY-520 (Array BioPharma, Boulder, CO, USA), AZD4877 (AstraZeneca, Cambridge, UK), MK0731 (Merck & Co., Kenilworth, NJ, USA), Litronesib (Kyowa Hakko Kirin Co., Ltd., Tokyo, Japan and Eli Lilly & Co., Indianapolis, IN, USA), ARQ 621 (ArQule Inc, Burlington, VT, USA), and 4SC-205 (4SC AG, Martinsried, Germany) [46]. Other KIF inhibitors [46], including GSK923295A (GSK plc, Brentford, UK) [47,48], are also in clinical development. Thus, investigational application of KIF26B to the setting of HO modulation has clinical precedent. Further studies using a clinically-relevant traumatic stimulus are required to both validate the mechanism of action of KIF26B and the effect of its inhibition on HO formation.

This study has several limitations. The sample size available for the HO-severity GWAS study was insufficient to give a meaningful examination of the genetic architecture of this trait. We also cannot exclude that any possible effect of the locus may not be upon *KIF26B* but upon another proximal or distal causal gene. The purpose of the association analysis here was exploratory to suggest possible candidate regions for closer scrutiny by other means. The finding of a sub-genome-wide association with KIF26B was complementary to other lines of evidence suggesting further functional exploration. We used the C2C12 myoblast cell line to model HO in vitro, as muscle precursor cells may contribute to the cellular origin of HO [49]. This is a sub-clone of a line originally derived from traumatized mouse muscle tissue [50,51], and will readily differentiate into myotubes, adipoblasts or osteoblasts when stimulated under appropriate conditions [19,37]. When stimulated with BMP2, this cell line is useful in the study of osteogenic differentiation and has been extensively used in the context of HO [24,37,52,53,54,55,56,57]. Other investigators have shown that the osteogenic media used here can result in dystrophic (non-apatitic) mineralization in other cell types, resulting in a false positive result using Alizarin Red S as the measure of a biological response [58,59]. In this study we showed that osteogenic media alone was insufficient to induce osteogenic differentiation of C2C12 cells in either the wild type or *Kif26b* CRISPR cells, as measured by Alizarin Red S or by RT-qPCR expression of *Runx2*, *Osx*, or *Bglap*. These osteogenic responses were induced with the addition of BMP2 to the osteogenic media in the wild type, but not the *Kif26b* CRISPR cells, confirming our hypothesis that KIF26B is necessary for BMP-induced osteogenic differentiation of C2C12 cells and that the biological response was osteogenic rather than simply dystrophic calcification.

Whilst the mechanistic insight from this model has been valuable, this approach also has limitations. It recapitulates a single cell of origin picture of acquired HO that is known to be complex and involving the interaction of multiple cell types of different embryonal lineages, and the dominant initiating cell type remains unknown. Whilst BMP2 is a reliable inducer of osseous trans-differentiation, it does not fully recapitulate the biology of a traumatic tissue injury.

In summary, we used a combination of cohort, bioinformatics, retrieval sample analysis and in-vitro experimental data to explore the biology of post-traumatic HO. Our data suggest a role for KIF26B in modulating disease severity. Further animal and clinical studies will be required to determine whether modulation of the associated signaling pathways may find clinical application in reducing the severity of post-traumatic HO.

## 4. Materials and Methods

### 4.1. Study Oversight and Populations

The GWAS cohort comprised 194 men and 216 women of UK European ancestry, mean age (±standard deviation) 63.4 ± 8.9 years, who had previously undergone hip replacement for idiopathic osteoarthritis and were genotyped as previously described [60]. Patients with a history of inflammatory arthritis, secondary arthritis due to trauma, avascular necrosis, developmental and childhood hip disorders, and subjects who had taken immuno-suppressant agents or bisphosphonates for a continuous period of greater than 6 months since primary THA were excluded from the study. All procedures were conventional cemented primary total hip replacements in patients who had no history of hip or pelvic surgery, and all prosthesis constructs used a metal or ceramic on conventional ultra-high molecular weight polyethylene bearing. All had HO on plain anteroposterior radiographs of the hip taken ≥1 year after surgery, and prior to any subsequent revision surgery. HO was graded by a single trained observer using the Brooker classification [61], as follows: small island(s) of bone (class 1, n = 201 subjects); bone spurs from pelvis and proximal femur leaving ≥1 cm between opposing surfaces (class 2, n = 133); bone spurs from pelvis and proximal femur leaving gap <1 cm (class 3, n = 69); apparent ankylosis of the hip (class 4, n = 1).

### 4.2. Screening GWAS

Standard GWAS quality control (QC) was conducted at the sample and variant levels and exclusions applied, as previously described [60,62]. Following QC, 410 subjects and 448,770 variants were imputed with IMPUTE2 [63] using the European reference panel (1000 Genomes Project, Dec 2010 phase I interim release) [64]. Variants with an imputation information score <0.4 and MAF <0.05 were excluded. A binomial association analysis of Brooker classes 1–2 versus classes 3–4 was undertaken on >10 million variants under the additive model and implemented in SNPTESTv2 [63]. The analysis was adjusted for age and sex as known risk factors for HO [65]. Data were pruned for linkage disequilibrium (LD) using the clumping function in PLINK [66]. The parameters used were: (a) significance threshold for index SNV 1 × 10^−5^, (b) LD threshold for clumping 0.20, and (c) physical distance threshold for clumping 500 kb. Statistical independence of the signals was also confirmed through conditional single-variant association analyses in SNPTESTv2.

### 4.3. Cell Culture

Human mesenchymal stem cells (hMSCs) were obtained from the bone marrow of three unrelated children undergoing osteotomy. Human multipotent adipose-derived stem cells (hMADS) were obtained from unrelated adults, and wild type (WT) C2C12 mouse myoblasts were purchased from Sigma-Aldrich (Burlington, VT, USA), passage 14 (cat. no. 91031101; lot no. 13K011). All cell types were cultured using standard protocols.

### 4.4. CRISPR-Cas9 Knockout

A CRISPR-Cas9 knockout kit (Origene, Rockville, MD, USA) for mouse *Kif26b* (SKU KN308785) was used to generate *Kif26b*-deficient cells. This targets the downstream part of *Kif26b* exon 1 and the start of exon 2, replacing the coding regions with a GFP-Puromycin cassette. The knockout of *Kif26b* transcript was verified using RT-qPCR and Western blotting.

### 4.5. Osteogenic Differentiation

MSCs and hMADS were seeded for 24 h in growth media, then 300 ng/mL human recombinant BMP2 was added for 48 h, then replaced with the osteogenic media. Osteogenic media contained 300 ng/mL BMP2, 10 mM β-Glycerophosphate, 10 nM Dexamethasone and 50 μg/mL Ascorbic Acid (Sigma-Aldrich). This osteogenic media, using 300 ng/mL BMP2 is well-established and widely used for trans-differentiation of C2C12 murine myoblasts [23,47]. Mouse C2C12 cells were cultured in T75 flasks in Dulbecco’s Modified Eagle’s Medium (DMEM) containing 4.5 g/L glucose and L-Glutamine (Lonza Group AG, Basel, Switzerland). An amount of 10% fetal bovine serum (FBS), 1% penicillin–streptomycin (P/S) and 1% L-Glutamine (L-G) were added to make the growth media. *Kif26b*^KO^ cells were cultured with 2 µg/mL puromycin. When passaging, cells were washed twice with phosphate buffered saline (PBS) before the addition of 0.25% trypsin-ethylenediaminetetraacetic acid (EDTA) to detach the cells. Cells were passaged every 48 h and split 1:5–1:10, confluence was avoided to prevent differentiation into myotubes. Cultures were routinely mycoplasma tested.

### 4.6. RNA Isolation and RT-qPCR

Total RNA was isolated using the Promega ReliaPrep™ RNA Cell, Tissue Miniprep System (Promega, Madison, WI, USA) and RNeasy UCP Micro Kit (Qiagen, Hilden, Germany). Reverse transcription of RNA was completed using the iScript™ cDNA Synthesis Kit (Bio-Rad Laboratories, Hercules, CA, USA). Triplicate technical repeats were conducted for each assay and normalized to a *β-Actin* housekeeping control in murine and *GAPDH* in human samples. Primers for SYBR green qPCR were designed using Primer-BLAST (NCBI; Appendix A). All primers were screened to avoid self-complementarity with Oligo Calc: Oligonucleotide Properties Calculator (http://biotools.nubic.northwestern.edu/OligoCalc.html, accessed on 9 January 2017). TaqMan gene expression probes (Invitrogen, Waltham, MA, USA) were used for mouse *Kif26b* and *β-Actin*.

### 4.7. Protein Isolation and Western Blotting

Cells were lysed in 0.1% RIPA-SDS lysis buffer with protease and phosphatase inhibitors. Rabbit anti-human KIF26B antibody was used for probing (ab121952, Abcam, Cambridge, UK), the immunized sequence identical in the mouse protein. Anti-phospho-SMAD 1/5/8 antibody (Cell Signaling Technologies, Danvers, MA, USA), anti-phospho-P38 (Cell Signaling Technologies) and anti-phospho-ERK1/2 (Cell Signaling Technologies) were used as recommended by the manufacturer. Immunocomplexes were visualized with ECL Western blotting detection reagent (GE Life Sciences, Amersham, UK) and bands were quantified using Image Studio Lite Ver 5.2 (Li-COR Biosciences, Lincoln, NE, USA).

### 4.8. Alizarin Red S Staining

Cells were washed twice with PBS and fixed overnight at 4 °C in 100% ethanol. Cells were then washed twice with PBS before the addition of 40 mM Alizarin Red S (Sigma-Aldrich), pH 4.2. The wells were washed extensively with 95% ethanol until all unbound stain was removed, the same number of washes was used for each well. Plates were air dried overnight and scanned for analysis. ImageJ Software (NIH: http://rsb.info.nih.gov/ij/, accessed on 1 May 2017) was used to quantify the percentage mineralized area in each well.

### 4.9. Immunocytochemistry

Cells were cultured in 24-well plates or on chamber slides and fixed in room temperature using 4% (*w*/*v*) paraformaldehyde (PFA)-PBS for 30 min. Anti-α-tubulin mouse monoclonal antibody was used to visualize microtubules (SC32293, Santa Cruz Biotechnology, Dallas, TX, USA). NL493 green donkey anti-mouse (NL009: R&D Systems, Minneapolis, MT, USA) secondary was used. Hoechst stain was used to stain the nuclei. Polyclonal rabbit anti-human KIF26B antibody (ab121952, Abcam) was used with anti-rabbit NL557 conjugated donkey IgG secondary (NL004: R&D Systems). Secondary antibody only was used for the control conditions, and settings were adjusted to remove non-specific binding.

### 4.10. Immunohistochemistry

Formalin-fixed, paraffin embedded rat lower limbs (a kind gift from L. Grover, Birmingham, UK) with previously-healed blade-induced Achilles tenotomy that had been performed according to the method described by Lin et al. [21], were dewaxed and rehydrated to water through a graded alcohol series. Slides were then washed in PBS, endogenous peroxidase activity quenched with 3% hydrogen peroxide for 10 min, washed in PBS and then incubated in 2.5% horse blocking serum (ImmPRESS anti-rabbit IgG reagent Kit, Vector Laboratories, Burlingame, CA, USA). Sections were then incubated with either a rabbit polyclonal anti-human KIF26B antibody (17422-1-AP, Proteintech, Rosemont, USA) or a rabbit polyclonal IgG isotype control antibody (ab37415, Abcam). Slides were incubated with the ImmPRESS Reagent before addition of ImmPACT DAB chromogen (SK-4105, Vector Laboratories). Sections were then counterstained with hematoxylin. Slides were scanned using a Panoramic 250 Flash III digital slide scanner (3D HISTECH, Budapest, Hungary) and images processed using QPath.

### 4.11. Statistical Analysis

Data are presented as mean ± SEM. Student’s *t*-test or, one-way or two-way ANOVA was used, with various post-hoc tests as indicated in the figure legends. All analyses were 2-tailed, and statistical significance was represented as *p* < 0.05. * *p* < 0.05, ** *p* < 0.01, *** *p* < 0.001, **** *p* < 0.0001. GraphPad Prism 7 (GraphPad software, San Diego, CA, USA) was used to present and analyze quantitative data.

## Figures and Tables

**Figure 1 ijms-23-09203-f001:**
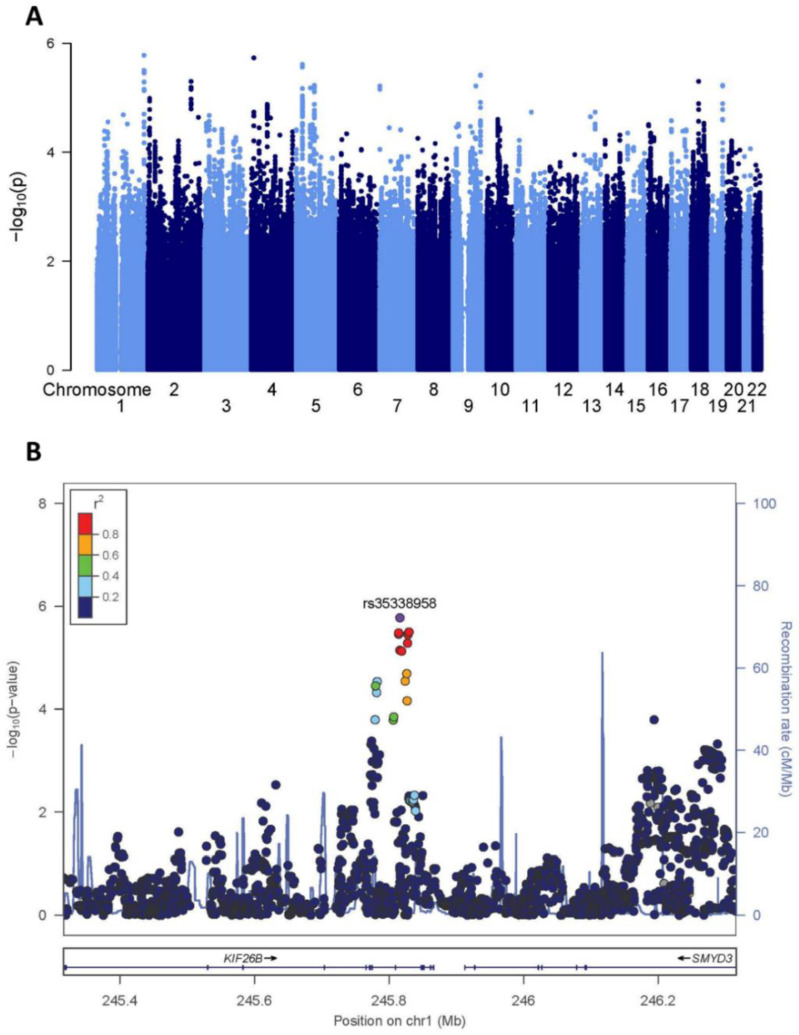
Screening genome-wide association analysis for HO severity-associated variants. (**A**) Manhattan plot showing the −log10 *p*-values for each variant (*y* axis) plotted against their respective chromosomal position (*x* axis). The horizontal dashed line denotes the genome-wide significance threshold *p* = 5 × 10^−8^. (**B**) Regional association plot of the *KIF26B* variant association with HO severity. Each filled circle represents the *p*-value of analyzed variants in the discovery stage plotted against their physical position (NCBI Build 37). The purple circle denotes rs35338958, which is the variant with the lowest *p*-value in the region. The colors of variants in each plot indicate their r^2^ with the lead variant according to a scale from r^2^ = 0 (blue) to r^2^ = 1 (red).

**Figure 2 ijms-23-09203-f002:**
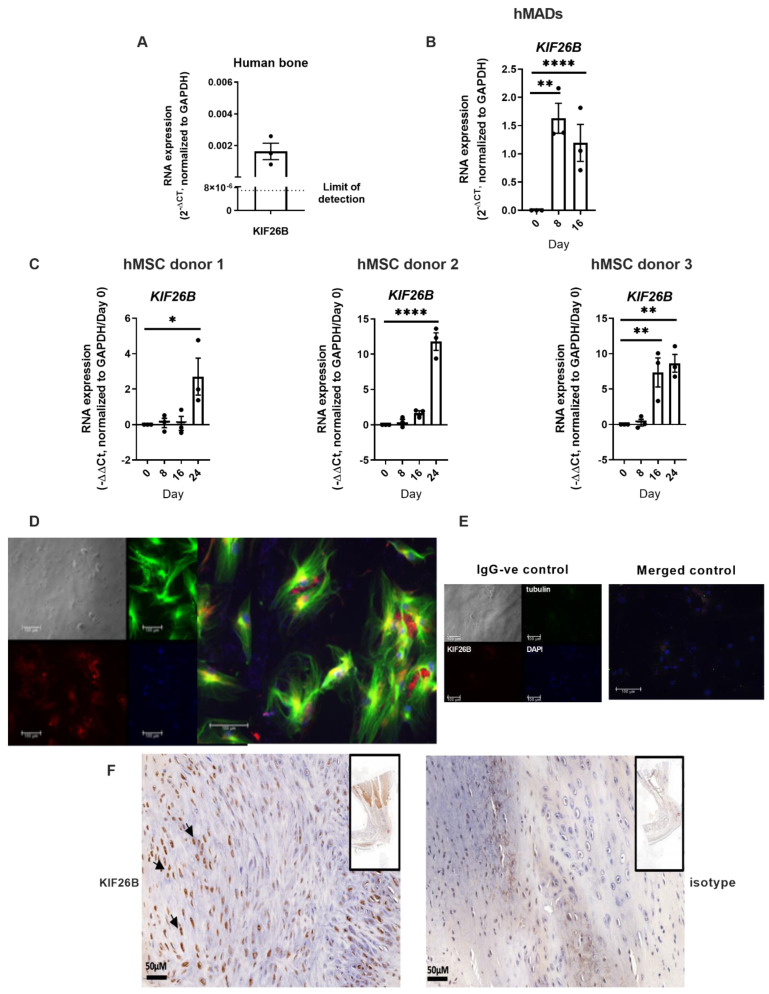
*KIF26B* is expressed in human bone and is induced in mesenchymal stem cells by BMP2 in vitro and in mammalian models of developing HO tissue. (**A**) RT-qPCR was used to measure the expression of *KIF26B* RNA in waste bone samples retrieved at joint replacement (n = 3 subjects). (**B**) *KIF26B* expression is induced in human multipotent adipose-derived stem cells (hMADs). RT-qPCR was used to measure the expression of *KIF26B* in hMADS at days 0, 8 and 16 of hMAD differentiation. Data were analyzed using 2^(−ΔCt)^ by normalizing to *GAPDH* (n = 3 biological replicates). Analyses were one-way ANOVA with Dunnett’s multiple comparisons; data are plotted as mean ± SEM. (**C**) *KIF26B* expression is induced in BMP2-stimulated human bone marrow derived mesenchymal stem cells (hMSC). RT-qPCR was used to measure the expression of *KIF26B* in hMSCs from 3 donors at days 0, 8, 16 and 24 of differentiation. Data were analyzed using −ΔΔCt, normalizing to day 0 and GAPDH (n = 3 biological replicate cultures). Analyses were one-way ANOVA with Dunnett’s multiple comparisons; data are plotted as mean ± SEM. (**D**) KIF26B protein is expressed in hMSCs under osteogenic conditions. Immunofluorescence staining of KIF26B (red), Tubulin (green) and DAPI (blue) in hMSCs (from 3 donors) differentiated for 8 days. (**E**) Secondary negative controls for immunofluorescence, left panel shows IgG isotype negative controls and right panel shows merged composite control image. (**F**) KIF26B protein is expressed in regions of new bone formation in post tendo-Achilles injury in rat. Left: anti-KIF26B antibody staining in regions of new bone formation in the rat Achilles tendon region; right: IgG control antibody staining. Arrows indicate the distinct nuclear localization of KIF26B (brown) to cells in left panel, including what appear to be hypertrophic chondrocytes. Inset for both panels is a lower magnification of the full section. Scale bar = 50 μM. * *p* < 0.05, ** *p* < 0.01, **** *p* < 0.0001.

**Figure 3 ijms-23-09203-f003:**
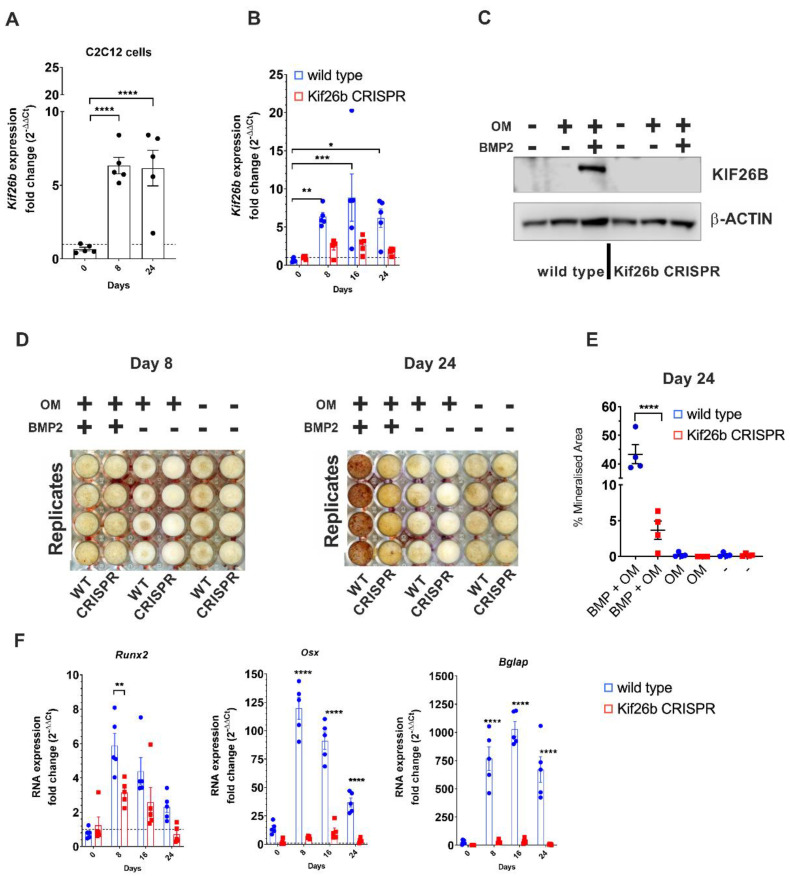
KIF26B is necessary for BMP-induced osteogenic differentiation of C2C12 murine myoblasts. (**A**) *Kif26b* mRNA expression is induced in C2C12 cells during BMP2 mediated differentiation. RT-qPCR was used to measure the expression of *Kif26b* at day 0 (confluency) relative to days 8 and 24 of differentiation when treated with 300 ng/mL BMP2 and osteogenic supplements. Technical replicates from a representative experiment (from 3 independent differentiations): each data point represents an individual well. Analyses were one-way ANOVA with Dunnett’s multiple comparisons; data are plotted as mean ± SEM. (**B**) Upregulation of *Kif26b* RNA is abrogated in *Kif26b^CRISPR^* C2C12 cells. RT-qPCR was used to measure the expression of *Kif26b* at days 8, 16 and 24 of differentiation when treated with 300 ng/mL BMP2 and osteogenic supplements relative to day 0 (confluency). Technical replicates from a representative experiment (from 3 independent differentiations): each data point represents an individual well. Analyses were one-way ANOVA with Dunnett’s multiple comparisons; data are plotted as mean ± SEM. (**C**) Upregulation of KIF26B protein is abrogated in *Kif26b^CRISPR^* C2C12 cells. KIF26B Western blot of WT and *Kif26b^CRISPR^* C2C12 myoblasts that were differentiated for 8 days in three different media. OM: media with osteogenic supplements. (**D**,**E**) In vitro mineralization is inhibited in *Kif26b^CRISPR^* C2C12 cells. (**D**) Alizarin Red S calcium staining of WT and *Kif26b^CRISPR^* C2C12 myoblasts after 8 and 24 days of differentiation. (**E**) Percentage mineralized area per well at day 24, as measured by percentage Alizarin Red S staining. Analyses were one-way ANOVA with Tukey’s post-hoc test. (**F**) Induction of osteogenic genes is prevented in *Kif26b^CRISPR^* C2C12 cells. Cells were cultured in media with osteogenic supplements and BMP2, and RNA levels of *Runx2, Osterix (Osx)* and *Osteocalcin (Bglap)* were measured by RT-qPCR. Analyses were two-way ANOVA with Sidak’s multiple comparisons. * *p* < 0.05, ** *p* < 0.01, *** *p* < 0.001, **** *p* < 0.0001.

**Figure 4 ijms-23-09203-f004:**
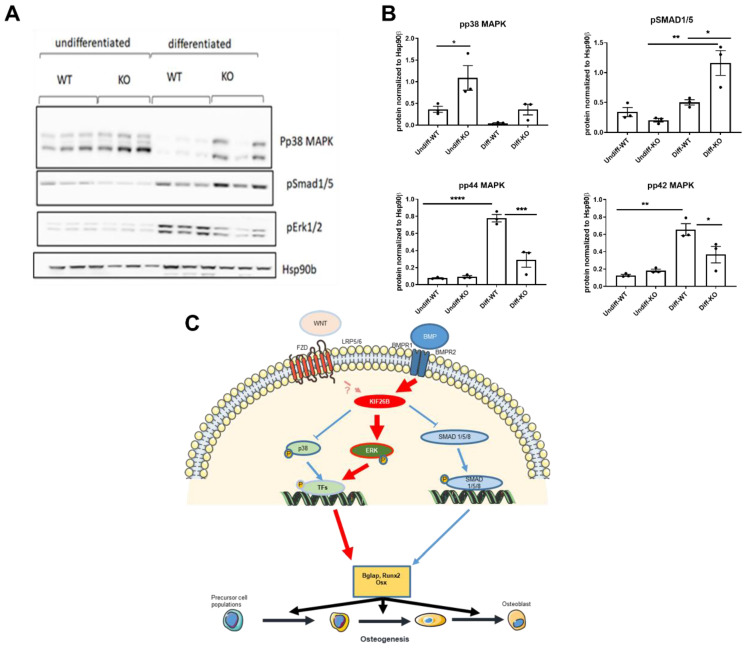
KIF26B-deficiency leads to dysregulated osteogenic signaling, impairing the activation of ERK MAPKs. (**A**,**B**) p38 MAPK, SMAD1/5 and ERK1/2 phosphorylation in WT and *Kif26b^CRISPR^* cells was assessed by Western blotting. Cells were differentiated for 8 days (WT/KO) (n = 3 biological replicates). (**B**) Western blot data were quantified using Image Studio Lite Ver 5.2. Analyses were one-way ANOVA with Tukey’s multiple comparison test. Data are plotted as mean ± SEM. (**C**) Proposed model for the molecular mechanism of KIF26B and CASC20 in osteogenesis. Both KIF26B and CASC20 expression is induced by BMP2. Whilst the mechanistic contribution of CASC20 to HO is unclear, KIF26B appears to control the balance of signaling pathways that collectively regulate then expression of osteogenic genes. * *p* < 0.05, ** *p* < 0.01, *** *p* < 0.001, **** *p* < 0.0001.

## Data Availability

All summary statistics will be deposited to the GWAS catalogue (https://www.ebi.ac.uk/gwas/, accessed on 1 July 2022) and the MSK portal (http://mskkp.org/, accessed on 1 July 2022) upon manuscript publication.

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
