# Peer review of "The Kinesin Gene KIF26B Modulates the Severity of Post-Traumatic Heterotopic Ossification"

_ijms, 2022, doi:10.3390/ijms23169203_

Round 1
Reviewer 1 Report
The manuscript “The kinesin gene KIF26B modulates the severity of post-traumatic heterotopic ossification” by George AE Pickering and colleagues addresses the clinically challenging problem of heterotopic ossification following trauma. The introduction in very brief and a little more background information could be helpful for the reader to understand the topic.
Abstract: The Abstract is written in a very general way and does not necessarily attract readers.
Methods:
1. GWAS: The results start with “As anticipated given the limited sample size…” – a cohort of over 400 patients seems quite large. Therefore, a bit more detailed information in the M&M section would help the reader to better understand the presented results.
2. The amount of rhBMP2 is very high (300 ng/ml).
3. Using 10 mM β-Glycerolphospate for osteogenic differentiation was shown to induce dystrophic mineralization or non-apatitic mineralization in cells generally incapable of osteogenic differentiation (L. Lammers, et al. 2012 Stem Cell Res / F. Langenbach and J Handschel 2013 Stem Cell Res & Ther). These formed crystals can be detected by Alizarin Red S or Von Kossa staining and will therefore affect the results.
4. How was the stability of the housekeeping gene tested between different individuals?
5. The description of the in vivo HO models (mouse and rat) are missing.
Results: Figure 3D: How can I be explained that there is less mineral formation in undifferentiated C2C12 cells when compared to osteogenically differentiated (OM) C2C12 cells?
Discussion: Parts of the discussion could be moved to the introduction, e.g. current therapies and comparison to FOP as this is not directly discussed with the presented results. Discussion should focus on discussing the results.
I’m not a native speaker. But for me the manuscript is sometimes hard to read. Considering most readers being non-native speakers – revision of the English might be helpful.
Reviewer 2 Report
The authors identify candidate molecular targets for severe HO. The combine the GWAS analysis with molecular biology approaches to confirm. They confirmed KIF26B as a target gene. Next they test if this gene is upregulated by BMP2 and also if the gene regulates osteogenesis related genes such as RUNX et. This is a very solid research publication, however the authors need to include negative controls n their result section. They need to include the negative secondary control images for IF. They need to confirm that crisper had no side effect on other genes and is specific to the KO.
Round 2
Reviewer 1 Report
abbreviations (e.g. FAPs, BMP, mTOR, RAR, etc.) shall be written out when first time used and/or a abbreviation list shall be provided.
Please refer to your own bioRxiv preprint:
Is the data part of a PhD thesis? Than this should be acknowledged.
Author Response
We thank the reviewer for these additional comments.
We have included in the introduction reference to the bioRxiv article (new reference 15)
The work is not part of a PhD thesis (although a PhD student has contributed in small part to some of the work).
Many thanks again, we are happy to address any further questions arising
Mark Wilkinson on behalf of all authors
Reviewer 2 Report
The authors addressed the concerns
Author Response
Thanks you for your time in reviewing this revised manuscript.
Best wishes
Mark Wilkinson, on behalf of all authors